# Withdrawal-Induced Delirium in Opioid Dependence: A Systematic Review

**DOI:** 10.3390/brainsci15101118

**Published:** 2025-10-17

**Authors:** Nikodem Świderski, Patryk Rodek, Krzysztof Kucia

**Affiliations:** 1Student Scientific Organization of Department and Clinic of Adult Psychiatry, Faculty of Medical Sciences, Medical University of Silesia, Ziolowa 45, 40-635 Katowice, Poland; 2Department and Clinic of Adult Psychiatry, Faculty of Medical Sciences, Medical University of Silesia, Ziolowa 45, 40-635 Katowice, Poland

**Keywords:** delirium, opioid withdrawal syndrome, chronic opioid use, opioid withdrawal delirium, antagonist-precipitated withdrawal, neuroadaptation

## Abstract

Background: Delirium is a rare but clinically significant complication of opioid withdrawal that remains poorly characterized in the literature. While classical withdrawal symptoms are well recognized, atypical presentations such as delirium are less frequently reported and often challenging to diagnose due to symptom overlap and heterogeneity of withdrawal syndromes. Methods: In this systematic review, we systematically analyzed available case reports and case series describing delirium precipitated by spontaneous opioid withdrawal, tapering, or antagonist-induced withdrawal. Twelve papers met inclusion criteria, comprising a total of fifteen case reports. Results: Most patients (*n* = 15) developed delirium within hours to days of withdrawal onset, often with fluctuating consciousness, disorientation, perceptual disturbances, and psychomotor changes. Reported risk factors included psychiatric comorbidity (major depressive disorder, anxiety disorder), concomitant use of psychotropic medication, rapid detoxification protocols, and potential exposure to adulterated substances. Management strategies varied but generally involved supportive care, benzodiazepines, antipsychotics, or reinstatement of opioid agonists. Conclusions: The findings highlight the need for heightened clinical awareness, careful differentiation from other withdrawal-related neuropsychiatric states, and systematic exclusion of organic etiologies. Despite the increasing number of patients affected by OWS, the knowledge available to date is based on case reports and a small case series, making it impossible to critically assess the prevalence or identify risk factors. Future research should aim to identify risk factors, optimize treatment, and explore novel diagnostic approaches, including AI-driven monitoring and connectomic analyses, to improve early detection and therapeutic outcomes in opioid withdrawal-associated delirium.

## 1. Introduction

Opioid withdrawal syndrome (OWS) is a well-recognized clinical entity arising from abrupt cessation or rapid tapering of opioid agonists in physically dependent individuals [1]. Its pathogenesis has been linked to neuroadaptations in the central nervous system (CNS) induced by a neurochemical imbalance caused by chronic administration of opioid receptor agonists [2]. The alterations mainly involve dopaminergic pathways, noradrenergic pathways and the endogenous opioid system, but are not limited to them [3,4,5]. Opioid cessation is followed by induced overactivation of the locus coeruleus (LC), hypoactivation of the mesocorticolimbic dopaminergic pathway and widespread autonomic dysregulation [6,7]. Clinically, OWS is characterized by rhinorrhea, lacrimation, dysphoria, myalgia, gastrointestinal hypermotility, diaphoresis, anxiety, and insomnia, but without the life-threatening potential typically associated with alcohol or benzodiazepine withdrawal [1,8].

The epidemiology of opioid use and misuse underscores the clinical relevance of OWS. Globally, over 60 million people report non-medical use of opioids, with the highest prevalence observed in North America, Eastern Europe, and parts of Asia [9,10]. Prescription opioid misuse, heroin, and synthetic opioids such as fentanyl contribute to a substantial public health burden, with rising rates of overdose and hospital admissions [11]. Chronic opioid exposure leads to physical dependence in a significant proportion of users, rendering withdrawal as well as its complications a frequent clinical scenario in both inpatient and outpatient settings.

Delirium is not considered a hallmark feature of OWS. It is defined by The Diagnostic and Statistical Manual of Mental Disorders, Fifth Edition (DSM-V) criteria as a sudden or fluctuating disturbance of consciousness and cognitive function that has a direct cause in a physiological condition resulting from another disease unit, intoxication, or withdrawal, often accompanied by perceptual disturbances, and altered psychomotor activity [12]. When delirium occurs in the context of opioid withdrawal, it is generally regarded as unusual and has been reported predominantly in the form of isolated case reports and small series. This rarity has led to under recognition, potential misdiagnosis (primary psychiatric disorder or intoxication), and delayed targeted management.

A particularly important subset is antagonist-precipitated withdrawal, where a full opioid antagonist (such as naloxone or naltrexone) is administered in the presence of residual agonist activity at opioid receptors [13]. In such cases, receptor occupancy shifts abruptly from agonist to antagonist, triggering a sudden and severe withdrawal state [14]. While the clinical picture is often dominated by intense autonomic and somatic symptoms, several published reports document the occurrence of delirium as a prominent or even leading manifestation [15]. With the growing use of opioid antagonists such as naloxone (for example, Narcan, a rapid-acting opioid overdose reversal agent available as a nasal spray) for overdose reversal, relapse prevention, and rapid detoxification, and their widespread availability among emergency services and first responders, understanding the risk of delirium during both spontaneous and antagonist-precipitated withdrawal has become a matter of practical clinical importance. The existing evidence base is sparse and fragmented across individual case descriptions.

This review aims to synthesize published case reports and small case series describing delirium during opioid withdrawal, including those precipitated by antagonist administration, highlighting clinical patterns, proposed mechanisms, and implications for prevention and management.

## 2. Materials and Methods

A comprehensive literature search was conducted in MEDLINE, EMBASE, Scopus, and Cochrane Library databases, using keywords and MeSH terms including “opioid withdrawal”, “delirium”, “heroin”, “fentanyl”, “antagonist”, “naltrexone”, “naloxone”, “buprenorphine” and “precipitated withdrawal” (Boolean terms used: “delirium” AND “opioid withdrawal” OR “heroin” OR “fentanyl” OR “antagonist” OR “naltrexone” OR “naloxone” OR “buprenorphine” OR “precipitated withdrawal”). Additional studies were identified by screening references in relevant articles. Inclusion criteria were (1) an article written in English, (2) publications from 1980 to August 2025, (3) the onset of delirium was in direct temporal relationship with opioid withdrawal, (4) withdrawal syndrome was the result of spontaneous withdrawal, precipitated-antagonist withdrawal, or induced by tapering the agonist dose, and (5) the authors described the clinical picture of individual patients and provided information on the therapeutic process undertaken.

Due to the small amount of data based on case reports and small case series, we waived the protocol registration, but nevertheless the correct structuring of the work was maintained.

The works selected for full text analysis were analyzed independently by two researchers, in case of disagreement on the inclusion of the work in the review, the decision fell to the third author supervising the work of the team.

Given the rarity of delirium during opioid withdrawal, only case reports and small case series were available, and no larger observational studies were identified. All reports included provided sufficient clinical detail to characterize the delirium and its management. Reports were considered regardless of whether delirium occurred spontaneously following opioid cessation or after administration of an opioid antagonist. Cases in which delirium could be attributed to alternative causes such as infection, metabolic disturbances, or concomitant alcohol or benzodiazepine withdrawal were noted but included only when opioid withdrawal was deemed the primary precipitant.

From each report, data were extracted regarding patient demographics, type of opioid use, mode of withdrawal, onset and duration of delirium, clinical features, management strategies, and outcomes. The extracted data were synthesized qualitatively, with emphasis on common clinical patterns, precipitating factors, management approaches, and patient outcomes, in order to identify trends and provide a practical overview for clinicians.

## 3. Results

The initial database search yielded 838 records. After removal of duplicates and abstract screening, 74 articles were retained for detailed full-text review. Of these, 12 publications met the inclusion criteria, comprising a total of 15 individual case reports of delirium occurring in the context of opioid withdrawal or antagonist-precipitated withdrawal (Figure 1). The quantitative data are shown in Table 1.

The majority of cases (*n* = 9) described delirium during spontaneous withdrawal from opioids such as heroin, opium, or methadone. In these reports, symptoms typically developed within 24–96 h of cessation or rapid tapering, and were characterized by fluctuating consciousness, disorientation, visual hallucinations, and psychomotor agitation. Three cases were associated with antagonist-precipitated withdrawal, most frequently involving naltrexone or naloxone administration in individuals with residual opioid agonist activity. These cases were notable for the abrupt onset of delirium, often within minutes to hours of antagonist exposure, and tended to present with severe autonomic hyperactivity alongside neuropsychiatric manifestations. Three additional cases were observed in the setting of opioid switching or iatrogenic withdrawal (during the detoxification process with tramadol and clonazepam or following inappropriate adjustment of buprenorphine maintenance therapy). Across all cases, management was primarily supportive, including close monitoring of vital signs, fluid therapy, and correction of metabolic disturbances. Pharmacological interventions were used in some instances: benzodiazepines were administered for agitation or anxiety, antipsychotics for severe psychotic features or hallucinations, and clonidine in selected cases to attenuate autonomic hyperactivity. In cases of iatrogenic or antagonist-precipitated withdrawal, reintroduction of opioid agonists or gradual tapering was employed, leading to resolution of delirium. Overall, all patients recovered fully without reported long-term neurological or psychiatric sequelae.

## 4. Discussion

### 4.1. Neuroadaptive Changes and Network Dysfunction in Opioid Withdrawal-Associated Delirium

Delirium is a complex and multifactorial neuropsychiatric syndrome characterized by acute disturbances in attention, awareness, and cognition, often accompanied by fluctuating levels of consciousness and perceptual distortions [26]. Its pathophysiology is highly heterogeneous, involving dysregulation of multiple neurotransmitter systems, including cholinergic hypofunction, dopaminergic and noradrenergic imbalance, and disturbances in GABAergic (γ-aminobutyric acid) and glutamatergic signaling [27]. Converging evidence also implicates neuroinflammation, oxidative stress, and hypothalamic–pituitary–adrenal (HPA) axis dysregulation, which collectively compromise cortical and subcortical network integrity [28,29].

During chronic opioid exposure, the central nervous system undergoes extensive neuroadaptive changes [30]. Prolonged stimulation of μ-opioid receptors leads to receptor downregulation and desensitization, altered G-protein–coupled signaling, and compensatory enhancement of excitatory neurotransmission, particularly in dopaminergic and noradrenergic pathways [31]. In addition, activation of opioid receptors causes inhibition of neuronal adenylyl cyclase (AC1) and a secondary decrease in cyclic adenosine 5′-monophosphate (cAMP) and protein kisane A (PKA). The adaptive increase in AC1 activity in the absence of the inhibitory action of opioids results in increased noradrenergic rebound activity during cessation and accounts for the autonomic symptoms of OWS [32] (Figure 2). Chronic exposure also induces homeostatic adjustments in cholinergic tonus and adaptations in glutamatergic and GABAergic pathways [33,34]. These mechanisms stabilize neural activity under continued opioid administration but simultaneously establish a state of latent vulnerability, rendering neural circuits highly sensitive to abrupt perturbations [35].

Abrupt discontinuation or rapid tapering of opioids leads to a breakdown of the established neuroadaptive mechanisms. The sudden loss of opioid-mediated inhibitory tone results in noradrenergic hyperactivity, dopaminergic dysregulation, and reduced cholinergic modulation [37]. Activation of the HPA axis and associated increases in cortisol further exacerbate this imbalance. In parallel, neuroinflammatory cascades, oxidative stress, and impaired synaptic plasticity contribute to destabilization of large-scale brain networks, amplifying cognitive dysfunction [38]. Antagonist-precipitated withdrawal represents an extreme version of this process, in which abrupt receptor blockade produces sympathetic overdrive, acute neurotransmitter imbalance, and profound cognitive disruption within minutes of administration [39,40].

Functional neuroimaging and connectomic research in delirium more broadly provides a framework for understanding these phenomena. Higher-order cognitive function relies on the dynamic interaction between large-scale brain networks, including the default mode network (DMN), executive network (EN), and salience network (SN) as well as a multitude of subcortical circuits [41]. The posterior cingulate cortex (PCC) and medial prefrontal cortex (mPFC) serve as a central hubs of the DMN, while the dorsolateral prefrontal cortex (DLPFC), dorsomedial prefrontal cortex (DMPFC), ventrolateral prefrontal cortex (VLPFC), pre-supplementary motor area (pre-SMA) and posterior parietal cortex (PPC) form critical nodes of the executive network, directing attention, working memory, and behavioral responses [42,43,44]. Under physiological conditions, the DMN and executive network exhibit anticorrelated intrinsic activity, ensuring adaptive allocation of attention and awareness [45]. In delirium, this anticorrelation is disrupted, leading to impaired environmental awareness and inappropriate behavioral responses [46] (Figure 3). Subcortical structures, including the intralaminar thalamic nuclei (ITN), mesencephalic tegmentum (MT), nucleus basalis (NBM), and ventral tegmental area (VTA), also show impaired functional connectivity during delirium episode. These regions, integral to the ascending reticular activating system, regulate arousal and consciousness [47,48]. Disruption of their interactions with cortical and striatal circuits likely contributes to the hallmark disturbances of consciousness and attention observed clinically.

Taken together, these findings suggest that opioid withdrawal delirium arises from the convergence of chronic neuroadaptive changes, abrupt neurochemical disbalance upon cessation, and network-level vulnerabilities that destabilize cognition and consciousness. The syndrome thus exemplifies how the breakdown of homeostatic adaptations in opioid dependence can propagate through neurotransmitter systems, inflammatory processes, and connectomic architecture to produce a clinical presentation of delirium (Figure 4).

The susceptibility to delirium in this context is further modulated by individual factors, including age, comorbid medical conditions, polypharmacy, and the presence of neurocognitive deficits [49,50]. Substance-specific factors, such as the potency of the opioid, chronicity of use, co-exposure to adulterants or contaminants, and abrupt administration of opioid antagonists, can precipitate more severe or rapid-onset presentations. Antagonist-precipitated delirium exemplifies this, with sudden receptor blockade inducing acute sympathetic overdrive, neurochemical disbalance, and profound cognitive disruption. These mechanistic insights suggest that opioid withdrawal delirium arises from a complex interplay of neuroadaptive, neurochemical, inflammatory, and network-level processes, which converge to destabilize brain function when homeostasis is abruptly perturbed.

### 4.2. Antagonist-Precipitated Delirium and Rapid Opioid Detoxification (ROD)

Several case reports underscore the phenomenon of antagonist-precipitated delirium, wherein administration of opioid antagonists, such as naloxone or naltrexone, triggers acute neuropsychiatric disturbances in individuals with chronic opioid exposure [51]. Typically, symptoms onset is rapid, occurring within minutes to a few hours post-antagonist administration, and the clinical presentation includes agitation, confusion, fluctuating attention, and hallucinations [52]. A related clinical scenario is rapid opioid detoxification (ROD), in which patients receive escalating doses of naltrexone (often combined with clonidine) under supervised inpatient protocols aimed at minimizing the duration of withdrawal [53]. In a retrospective study by Golden et al., five out of twenty consecutive methadone-maintained patients developed delirium meeting DSM-IV criteria during such a protocol [54]. This study was excluded from our case-based synthesis due to the absence of detailed clinical narratives describing the course of delirium in individual patients; nonetheless, it provides compelling evidence that ROD can precipitate clinically significant central dysfunction. Importantly, Golden and colleagues noted that delirium occurred more frequently in individuals with co-occurring psychiatric disorders, particularly those receiving SSRIs (selective serotonin reuptake inhibitors) or benzodiazepines, suggesting that pharmacological interactions or baseline neurobiological vulnerability may heighten risk. No clear associations with age or sex were identified. Moreover, affected patients demonstrated lower retention rates in treatment and higher risk of relapse, underlining the potential long-term clinical consequences of this complication. However, the study was limited by its small sample size and lack of systematic delirium assessment, which introduces a substantial risk of bias. The restricted cohort makes it difficult to generalize the findings to the broader population of opioid-dependent patients, and the absence of standardized clinical correlates of delirium reduces interpretability.

These findings are consistent with observations by Bell and Young, who examined outcomes of rapid detoxification protocols and reported a similarly elevated risk of delirium [55]. In their cohort, 60% of affected patients required short-term hospitalization, underscoring the clinical severity of the syndrome. Although the authors did not specify the exact duration of delirium, they observed that clinical symptoms emerged within approximately one hour of antagonist administration and persisted well beyond 6–8 h. Comparable to Golden’s report, these cases were also associated with reduced treatment retention and higher relapse rates, further reinforcing the notion that antagonist-precipitated delirium not only represents an acute clinical challenge but may also compromise long-term therapeutic outcomes.

There also exist more accelerated detoxification protocols, such as ultra-rapid opioid detoxification (UROD), in which opioid antagonists are administered under general anesthesia or deep sedation to precipitate and complete withdrawal within a matter of hours [56]. While such approaches aim to minimize subjective withdrawal distress, the use of anesthesia itself introduces additional physiological stressors, including cardiorespiratory instability and altered neurocognitive recovery, which may theoretically increase vulnerability to acute neuropsychiatric complications. Despite these considerations, no published clinical data are available on the risk of delirium specifically associated with UROD, and no case reports documenting such events have been identified. This absence of evidence leaves uncertainty regarding the true safety profile of ultra-rapid protocols in relation to delirium, highlighting the need for systematic investigation of both cognitive and behavioral outcomes in this setting.

The precise neurobiological underpinnings of antagonist-induced delirium remain incompletely understood [57]. At the most fundamental level, abrupt blockade of μ-receptors disrupts inhibitory control over dopaminergic and noradrenergic neurons, leading to hyperexcitability and profound dysregulation of arousal systems [58]. This is compounded by loss of compensatory cholinergic and glutamatergic adaptations established during chronic opioid exposure [59]. Emerging evidence further suggests that naloxone active metabolite 6β-naloxol may interact with κ-opioid receptors (KOR) [60]. Stimulation of these receptors exerts a distinct neurochemical effect compared to μ-receptor antagonism: KOR activation has been shown to selectively inhibit dopaminergic neurons in the VTA projecting to the mPFC [61,62,63]. The dopaminergic mesocortical pathway is regarded as critical for maintaining cognitive control, attention, and working memory, and its suppression can lead to dysphoria, perceptual disturbances, and deficits in executive functioning [64,65,66]. In the context of antagonist-precipitated withdrawal, such selective inhibition may synergize with the loss of μ-opioid tone, amplifying disruption of prefrontal network activity and thereby exacerbating the clinical picture of delirium.

Taken together, these findings suggest that precipitated withdrawal, whether induced deliberately in detoxification protocols or inadvertently in the context of emergency reversal, can overwhelm neuroadaptive homeostasis and destabilize brain network function. Although antagonist-precipitated delirium often resolves with supportive care, its abrupt and severe presentation underscores the importance of careful patient selection, slow titration, and intensive monitoring when initiating antagonist therapy [67]. Moreover, the possibility of receptor-specific interactions beyond simple μ-blockade, particularly through KOR-mediated inhibition of mesocortical dopamine transmission, highlights the need for mechanistic studies to clarify how sudden pharmacological perturbation of opioid systems translates into the complex neuropsychiatric picture of delirium.

### 4.3. Heterogeneity of Opioid Withdrawal Syndrome (OWS)

Opioid withdrawal typically presents with a multitude of autonomic, somatic, and neuropsychiatric symptoms. Classic manifestations include yawning, lacrimation, rhinorrhea, sweating, piloerection, gastrointestinal distress, myalgias, and insomnia, alongside affective symptoms such as irritability, anxiety, and dysphoria [68,69,70]. Beyond this prototypical presentation, withdrawal can manifest in more atypical or severe neuropsychiatric forms, including transient psychotic episodes, hallucinations, catatonia-like states, or hypoactive presentations characterized by lethargy, apathy, and psychomotor retardation [71,72,73]. Case reports have described also patients developing acute confusion, persecutory delusions, or severe agitation or even a set of symptoms that are part of neuroleptic malignant like syndrome [74,75,76].

This variability in symptomatology poses a significant challenge in recognizing delirium within the opioid withdrawal context. Many withdrawal symptoms, such as restlessness, insomnia, agitation, mood lability, and perceptual disturbances, are overlapping with the diagnostic criteria for delirium or could be misattributed to primary psychiatric disorder.

Structured assessment tools are essential for systematically evaluating both opioid withdrawal and delirium, given the overlapping clinical features. The Clinical Opiate Withdrawal Scale (COWS) is the most widely used instrument for assessing opioid withdrawal severity [77]. It comprises 11 items covering autonomic signs (sweating, lacrimation, piloerection, gastrointestinal upset), subjective symptoms (anxiety, irritability, restlessness, bone or muscle aches), and observable behaviors (tremor, yawning, pupil size). Each item is scored on a standardized scale, allowing clinicians to classify withdrawal severity from mild to severe, monitor progression over time, and guide pharmacologic intervention. Despite its utility, COWS primarily captures classical withdrawal features and may underrepresent atypical neuropsychiatric presentations such as catatonia or hallucinations [78]. For delirium assessment, the Confusion Assessment Method (CAM) and its intensive care adaptation (CAM-ICU) are widely validated tools [79,80]. CAM evaluates four key features: (1) acute onset and fluctuating course, (2) inattention, (3) disorganized thinking, and (4) altered level of consciousness. A diagnosis of delirium requires the presence of features 1 and 2, plus either 3 or 4 [81]. CAM-ICU modifies the assessment for non-verbal or mechanically ventilated patients, using visual and motor tasks to assess attention and cognition [82]. These tools are sensitive and relatively quick to administer, but their effectiveness may be limited in opioid withdrawal due to overlapping symptoms such as agitation, anxiety, or sleep disturbances, which can confound scoring however using both COWS and CAM concurrently allows clinicians to differentiate between pure withdrawal phenomena and emergent delirium, particularly in complex cases involving atypical presentations or rapid tapering. Incorporating repeated, longitudinal assessments is recommended, as both withdrawal severity and delirium symptoms fluctuate over time, and early detection of delirium can prevent progression to more severe complications.

In addition to careful assessment with COWS and CAM, it is critical to systematically rule out other organic causes of delirium, as opioid withdrawal can unmask or coexist with other medical conditions that independently precipitate acute cognitive changes [83]. Common contributors include metabolic disturbances (hypoglycemia, hyponatremia, hepatic or renal dysfunction), infections (urinary tract infections, pneumonia, sepsis), hypoxia or hypercapnia, acute cerebrovascular events, intracranial pathology (hemorrhage, tumor), and intoxications with other substances (alcohol, benzodiazepines, or stimulants) [84,85,86,87,88,89,90]. Cardiac, pulmonary, and endocrine disorders can further complicate the clinical picture [91,92]. Laboratory evaluation, vital sign monitoring, and neuroimaging when indicated are essential to exclude these alternative etiologies. Only after a thorough workup can delirium be attributed primarily to opioid withdrawal, allowing appropriate treatment planning and reducing the risk of overlooking life-threatening conditions. Together, the variable presentations of withdrawal, overlapping symptomatology, and potential coexisting organic etiologies underscore the diagnostic challenge of delirium in this context, necessitating careful, structured assessment and vigilance for atypical manifestations.

Correct recognition of OWS-induced delirium can have a significant impact not only on the therapeutic process of patients affected by this complication, but also shorten the course of hospitalization, reduce healthcare costs by correct distribution of both medical personnel and therapeutic methods used. In light of the increased prevalence of OWS and the use of opioid receptor antagonists not only in addiction medicine, but also in emergency medicine, awareness of the possible clinical presentation of OWS in the form of delirium, as well as other uncharacteristic neuropsychiatric disorders, is crucial to the appropriate therapeutic process.

### 4.4. Future Directions: AI-Based Approaches in Opioid Withdrawal Delirium

Detecting delirium in the setting of opioid withdrawal poses a unique challenge, not only because its presentation overlaps with classical withdrawal symptoms, but also because clinical manifestations may fluctuate rapidly and remain under-recognized until complications arise. Conventional bedside assessments, although valuable, are often insufficient to capture early or subtle changes, particularly in heterogeneous populations with co-occurring psychiatric or somatic conditions. This diagnostic uncertainty underscores the need for more precise tools capable of continuously monitoring brain and behavioral states. Recent advances in artificial intelligence (AI) have opened promising avenues in this regard. Predictive models trained on large-scale electronic health records have already demonstrated strong performance in identifying delirium across critical care and perioperative populations, with reported area under the receiver operating characteristic curve (AUROC) between 0.77 and 0.89 [93,94]. Parallel developments are emerging in real-time monitoring. Portable EEG-based platforms, enhanced by deep learning, have been used to identify electrophysiological signatures of delirium within minutes, enabling bedside application even outside ICU settings. Continuous EEG combined with deep neural networks has further demonstrated the ability to track fluctuations in consciousness with good sensitivity and specificity, suggesting potential for proactive rather than reactive management [95]. Moreover, multimodal approaches that integrate neuroimaging, electrophysiology, and environmental data (ambient light, circadian disruption, noise exposure) have been piloted with encouraging predictive performance [96]. At the same time, ethical considerations must not be overlooked. Bias in training datasets may lead to systematic underdiagnosis or misclassification in vulnerable subgroups, reinforcing existing disparities.

In summary, future research on opioid withdrawal-related delirium should move beyond descriptive case reports and incorporate network-level neuroscience with AI-powered detection systems. Combining fMRI, EEG, wearable biosensors, and machine learning applied to clinical records could provide individualized risk stratification, early biomarkers of vulnerability, and optimized treatment strategies. Such approaches have the potential not only to improve diagnostic accuracy but also to enhance treatment retention and reduce relapse, ultimately translating into a precision medicine framework for one of the most challenging complications of opioid withdrawal.

### 4.5. Risk of Bias

The main limiting factor was the reliance of the analysis entirely on case reports and small case series, which makes it impossible to extrapolate results to populations and increases the risk of systematic errors. Some of the studies did not systematically analyze patients using specific tests, instead relying on the clinical interpretation of the patient’s condition by the subjective opinion of the treating physician, which could vary in detail and assessment of the clinical condition. The lack of prospective controlled clinical trials makes it impossible to reliably estimate the prevalence or specific risk factors for delirium induced-OWS. In addition, publication bias may favor the reporting of atypical or particularly severe cases, further biasing the available data. In conclusion, these results should be interpreted with caution, while they underscore the urgent need for prospective clinical trials.

## 5. Conclusions

In conclusion, delirium precipitated by opioid withdrawal or rapid antagonist-induced detoxification represents a rare but clinically significant phenomenon, arising from a complex interplay of neuroadaptive, neurochemical, inflammatory, and network-level processes. Chronic opioid exposure induces extensive neuroadaptations, creating latent vulnerability that can be unmasked by abrupt cessation or rapid receptor blockade, leading to dysregulated neurotransmission, heightened sympathetic activity, and impaired cognitive function. The clinical heterogeneity of opioid withdrawal, encompassing classical autonomic symptoms as well as atypical presentations such as catatonia or psychosis, poses substantial diagnostic challenges, often masking the onset of delirium. Despite the existence of case reports in the literature describing delirium associated with opioid withdrawal syndrome, further research is urgently needed to clarify the specific risk factors, optimize therapeutic strategies, and develop predictive models for treatment efficacy and relapse risk. Such studies would be critical to improve clinical outcomes and provide evidence-based guidance for clinicians managing opioid-dependent individuals undergoing withdrawal or detoxification protocols.

## Figures and Tables

**Figure 1 brainsci-15-01118-f001:**
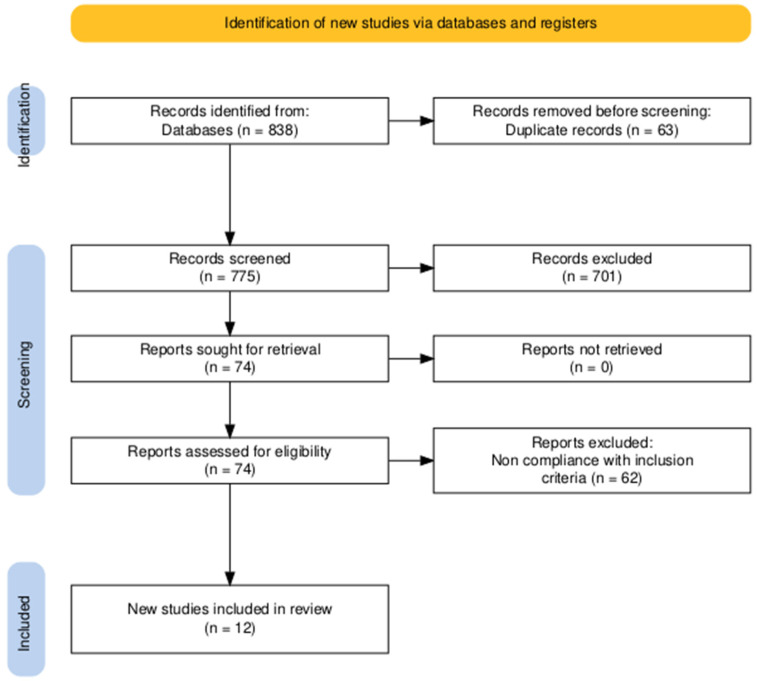
PRISMA flow-chart.

**Figure 2 brainsci-15-01118-f002:**
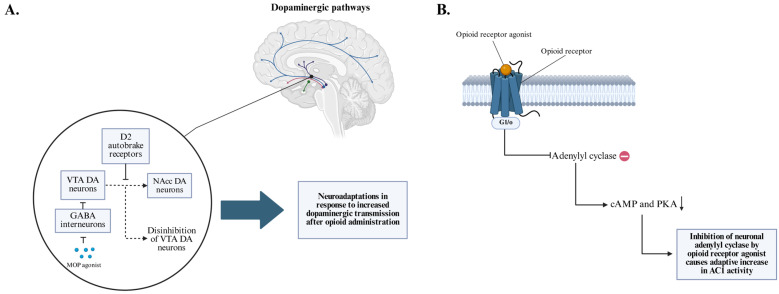
(**A**) Opioid receptor agonists inhibit GABAergic interneurons, disinhibiting VTA dopaminergic neurons and inducing adaptive changes in dopamine receptor sensitivity, clinically related to the neurobiological basis of addiction development. (Figure adapted from another of the authors’ own work [36]). (**B**) Simultaneously, receptor activation suppresses adenylyl cyclase activity, with compensatory AC1 upregulation driving noradrenergic hyperactivity during spontaneous withdrawal (GABA—γ-aminobutyric acid; VTA DA neurons—ventral tegmental area dopaminergic neurons; NAcc DA neurons—nucleus accumbens dopaminergic neurons; AC1—neuronal adenylyl cyclase; cAMP—cyclic adenosine-5′-monophosphate; PKA—protein kinase A).

**Figure 3 brainsci-15-01118-f003:**
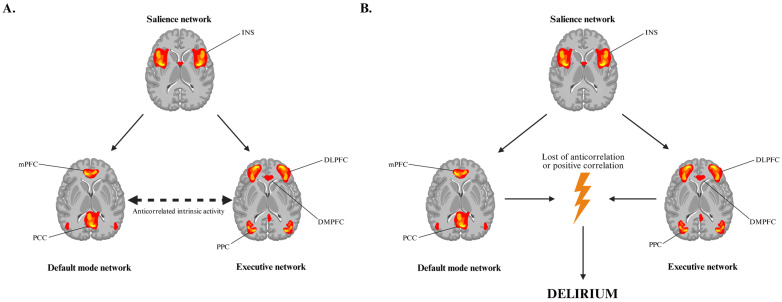
(**A**) Under physiological conditions, the executive and default mode networks function in anticorrelation, with their intrinsic activity dynamically regulated by the salience network to redirect attention according to external stimuli. (**B**) During delirium, fMRI reveals reduced anticorrelation or even positive correlation between these networks. These changes are postulated to be the cause of impaired consciousness, attention and voluntary motor functions during an episode of delirium (mPFC—medial prefrontal cortex; PCC—posterior cingulate cortex; DLPFC—dorsolateral prefrontal cortex; DMPFC—dorsomedial prefrontal cortex; PPC—posterior parietal cortex; INS—insula).

**Figure 4 brainsci-15-01118-f004:**
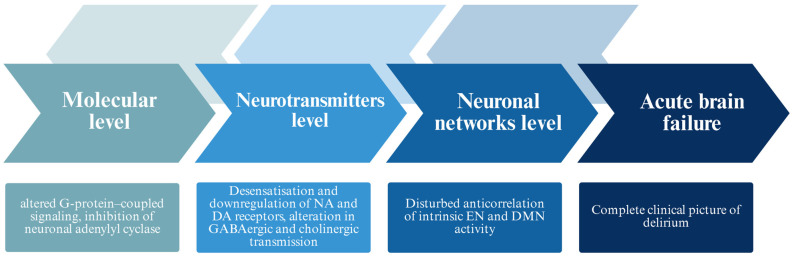
Simplified schematic of proposed levels of alterations contributing to opioid withdrawal–induced delirium. The diagram illustrates how changes at each level may interact nonlinearly, with bidirectional influences, ultimately affecting higher-order CNS functions that determine consciousness and cognitive performance. Relationships are simplified for clarity and do not represent all possible pathways or feedback mechanisms.

**Table 1 brainsci-15-01118-t001:** Works included in the final synthesis of this review.

Author(s) (Year)	Age/Sex	Opioid	Cessation Sequence	Onset Time of Delirium	Clinical Presentation	Treatment	Outcomes
Singh & Parker (2022) [16]	20/M	Non specified chronic opioid use	Opioid withdrawal (in general)	2 days	Psychotic manifestations (delusions, auditory pseudohallucinations), agitation, prosopagnosia, sleep disturbances	10 mg/day lorazepam, 10 mg/day haloperidol, discharged on 50 mg naltrexone per day	Complete resolution
Pandey et al. (2024) [17]	23/M	Heroin	Opioid withdrawal (in general)	5 days	Severe agitation, sleep disturbances, disturbance of consciousness	2 mg/h midazolam, 32 μg/h dexmedetomidine, discharged on 50 mg naltrexone per day	Complete resolution in 24 h
Aggarwal et al. (2011) [18]	23/M	Heroin	Opioid withdrawal (in general)	4 days	Fully disoriented, agitation, prosopagnosia, irrelevant speech, sleep disturbances	200 mg/day tramadol, 3 mg/day clonazepam, discharged on 50 mg naltrexone per day	Complete resolution in 2 weeks
Raj et al. (2017) [19]	25/M	Codeine and heroine	Opioid withdrawal (in general)	7 days	Psychotic manifestations (delusions, multimodal hallucinations), agitation, prosopagnosia, irrelevant speech, sleep disturbances	Initiatively 5 mg haloperidol, 4 mg lorazepam, 0.1 mg clonidine, 15 mg diazepam, then 1.5 mg/day clonidine and 8 mg/day lorazepam, discharged on 50 mg naltrexone per day	Complete resolution in 7 days, lapsed 3 times and was readmitted with similar presentation
Sharma et al. (2017) [20]	56/M	Opium	Detoxification protocol	2 days	Severe agitation, irrelevant speech, sleep disturbances	Initiatively 8 mg lorazepam, 5 mg haloperidol, 50 mg promethazine, then 300 mg/day tramadol and 1 mg/day clonazepam, discharged on 50 mg/day naltrexone and 50 mg/day quetiapine	Complete resolution in 58 h
Sharma et al. (2017) [20]	38/M	Opium	Opioid withdrawal (in general)	2 days	Confusion, agitation, stereotypical movements, sleep disturbances	200 mg/day tramadol, 1.5 mg/day risperidone, 3 mg/day clonazepam, discharged on 50 mg/day quetiapine	Complete resolution in 48 h
Das et al. (2017) [21]	26/M	Heroin	Substitution therapy	5 days	Confusion, agitation, disorientation, sleep disturbances	8 mg/day buprenorphine, 2 mg/day naloxone	Complete resolution in 48–72 h
Das et al. (2017) [21]	29/M	Heroin	Substitution therapy	4 days	Confusion, agitation, disorientation, sleep disturbances	8 mg/day buprenorphine, 2 mg/day naloxone	Complete resolution in 48–72 h
Aggarwal et al. (2024) [22]	24/M	Non specified chronic opioid use	Precipitated opioid withdrawal	Same day	Psychotic manifestations (delusion, visual hallucinations), irrelevant speech, agitation, disorientation	Initiatively 70 mg diazepam, 17.5 mg haloperidol, then 15 mg/day diazepam and 1.5 mg/day clonidine	Complete resolution in 7 days
Miles et al. (2020) [23]	45/M	Non specified chronic opioid use	Precipitated opioid withdrawal	Same day	Severe agitation	2 mg lorazepam	Complete resolution in 2 days
Parkar et al. (2006) [24]	4 patients aged 20–38 years	Opium	Opioid withdrawal (in general)	3–7 days	Not specified, the diagnosis of delirium made by the attending physician	Not specified	Complete resolution
Hanna & Swetter (2018) [25]	70/M	Chronic alcohol and opioid use	Precipitated opioid withdrawal	1 day	Severe agitation, disorientation	1 mg clonidine, 2 mg lorazepam, methadone	Complete resolution in 7 days

## Data Availability

No new data were created or analyzed in this study.

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
