# Peer review of "Withdrawal-Induced Delirium in Opioid Dependence: A Systematic Review"

_brainsci, 2025, doi:10.3390/brainsci15101118_

Round 1

Reviewer 1 Report

Comments and Suggestions for Authors

This manuscript offers a comprehensive narrative synthesis of delirium occurring during opioid withdrawal, supported by a structured review of published case reports and small case series. It addresses a clinically under-recognized but increasingly relevant issue, especially in the context of growing use of opioid antagonists in various settings.

Strengths:

  • The introduction provides excellent background and justification for the review.

  • The methodology is transparent, with clear inclusion criteria and comprehensive database coverage.

  • The discussion provides an in-depth, mechanistically informed interpretation, integrating neuroadaptation, neuroinflammation, and network dysfunction, supported by recent literature.

  • The inclusion of AI-based future perspectives is innovative and forward-looking.

Suggestions for improvement:

  1. Methods:

    • While the search strategy is well described, it would be useful to include the exact search strings (e.g., in Supplementary Materials) and PRISMA checklist to strengthen reproducibility.

    • Clarify how disagreements in study selection were resolved (e.g., between reviewers).

  2. Results:

    • The case synthesis table is informative but dense; consider improving readability by standardizing terminology (e.g., “precipitated withdrawal” vs. “antagonist-induced”).

    • Explicitly summarize key patterns (e.g., average time to onset, duration, treatment strategies) in a short paragraph following the table.

  3. Discussion:

    • The discussion is strong but could benefit from a brief paragraph summarizing clinical implications for emergency medicine, addiction medicine, and psychiatry.

    • Consider explicitly stating the limitations: reliance on case reports, potential publication bias, heterogeneity of reporting, and absence of standardized delirium assessments in some cases.

  4. Figures:

    • The conceptual figures (especially Figures 2–4) are excellent. It may be helpful to add short legends or arrows indicating clinical relevance (e.g., “precipitated withdrawal” vs. “spontaneous withdrawal”).

Overall, this is a well-prepared and insightful manuscript that makes a valuable contribution to the field. With minor adjustments to methodology description and clearer clinical implications, it will be ready for publication.

Author Response

Thank you very much for your detailed review and the suggestions you sent. We believe that the suggested changes will significantly affect the merits and overall perception of the work.

Below I am sending bulleted responses to the submitted comments:

Comment 1: 

While the search strategy is well described, it would be useful to include the exact search strings (e.g., in Supplementary Materials) and PRISMA checklist to strengthen reproducibility.

Clarify how disagreements in study selection were resolved (e.g., between reviewers).

Response 1: 

In the materials and methods section, the search phrases used in the database analysis have been added, along with the exact Boolean expression used in the search process (LINE 85-87). The completed PRISMA-checklist was also sent.

Clarification of contentious situations in the work selection process has been added (LINE 94-96)

Comment 2:

The case synthesis table is informative but dense; consider improving readability by standardizing terminology (e.g., “precipitated withdrawal” vs. “antagonist-induced”).

Explicitly summarize key patterns (e.g., average time to onset, duration, treatment strategies) in a short paragraph following the table.

Response 2:

Table 1. has been modified, in particular the column of the opioid cessation regimen (LINE 117).

A short paragraph summarizing the contents of the table and behind this included work is directly below the table (LINE 119-131).

Comment 3: 

The discussion is strong but could benefit from a brief paragraph summarizing clinical implications for emergency medicine, addiction medicine, and psychiatry.

Consider explicitly stating the limitations: reliance on case reports, potential publication bias, heterogeneity of reporting, and absence of standardized delirium assessments in some cases.

Response 3:

The paragraph was added as suggested (LINE 360-367).

A subsection summarizing the total risk of bias was added to the discussion (LINE 403-413).

Comment 4:

The conceptual figures (especially Figures 2–4) are excellent. It may be helpful to add short legends or arrows indicating clinical relevance (e.g., “precipitated withdrawal” vs. “spontaneous withdrawal”).

Response 4:

We sincerely thank you for the feedback you sent.
Information about the clinical significance of the molecular and functional changes described has been added to the legends of Figures 2-3.

Once again, we are grateful for the comments sent and are open to feedback on the changes made.

Reviewer 2 Report

Comments and Suggestions for Authors

The authors performed a systematic review of the opioid withdrawal induced delirium (spontaneous vs tapering vs antagonist-induced withdrawal. The article is well written, and methodology is scientifically sound. In particular, the discussion was well written especially the underlying neuropathology.

The authors mentioned that " Delirium is not considered a hallmark feature of OWS", which is true. In general, Delirium a common waning and waxing level of consciousness is a particular concern in ICU, secondary to environment, medications especially BZDS. 

The case reports mentioned onset of delirium although some of them are non-specific eg: Parker et all. The reported case reports which mention about the onset after opioid reversal such as Hanna & Swetter et all , Aggarwal et al, Das et al strongly support to the authors hypothesis of opioid reversal induced delirium. In the rest of the case reports some of the patients have underlying psychiatric disorders which makes it hard to relate the opioids are the only contributors of withdrawal.

Neuro-adaptative changes were well written and also the role of AI in the future

Reviewer 3 Report

Comments and Suggestions for Authors

1-) please mention the number of them.

. Most patients developed delirium within hours to days of withdrawal 18
onset, often with fluctuating consciousness, disorientation, perceptual disturbances, and 19
psychomotor changes. 

2-)please give more information about comorbidity.  for example which psychiatric comorbidity

Reported risk factors included psychiatric comorbidity, concomi- 20
tant use of psychotropic medication, rapid detoxification protocols, and potential expo- 21
sure to adulterated substance

3-)please make this sentence specifically related to your study.

Despite increasing recognition, robust clinical studies are lacking, and knowledge 26
remains restricted to individual cases.

4-)please make sure that selected studies are relevant.

5-)please add more information related to guideliness followed in the review.

6-)if possible please improve figure quality.

7-)please be sure the scientific information in the figurs are accurate.

8-)please be sure the figure 2 is relevant to your study.

9-)please mention if chatbots are utilized for the preparation of the manuscripts.

Author Response

We sincerely thank you for your thorough review of our work and the comments you sent. We are confident that the changes undertaken will positively affect both the merits of the work and the overall reception.

I am sending the bulleted responses to your comments below:

Comment 1:

please mention the number of them.

Most patients developed delirium within hours to days of withdrawal 18
onset, often with fluctuating consciousness, disorientation, perceptual disturbances, and 19
psychomotor changes. 

Response 1:

The number of patients for the sentence has been added (LINE 18)

Comment 2:

please give more information about comorbidity.  for example which psychiatric comorbidity

Reported risk factors included psychiatric comorbidity, concomi- 20
tant use of psychotropic medication, rapid detoxification protocols, and potential expo- 21
sure to adulterated substance

Response 2:

Psychiatric comorbidities that correlated with the occurrence of OWS-induced delirium in the studies cited in the discussion were listed (LINE 21)

Comment 3:

please make this sentence specifically related to your study.

Despite increasing recognition, robust clinical studies are lacking, and knowledge 26
remains restricted to individual cases.

Response 3:

The sentence has been edited (LINE 26-29)

Comment 4: 

please make sure that selected studies are relevant.

Response 4: 

The included works were reanalyzed independently by two researchers. A decision was made not to exclude additional papers, and the total number of papers is 12.

Comment 5: 

please add more information related to guideliness followed in the review.

Response 5:

The Materials and Methods section has been enriched with a section on a contentious situation regarding the inclusion of a given paper in a review (LINE 96-98), and the search phrases used in the databases are provided, along with the exact boolean expressions (LINE 87-89).
In addition, a PRISMA checklist was submitted.

Comment 6:

if possible please improve figure quality.

Response 6: 

Figures at DPI 600, the maximum resolution possible, were submitted with the manuscript.
We suspect that the blurred appearance of the figures is the result of image processing by the WORD file.

Comment 7:

please be sure the scientific information in the figurs are accurate.

Response 7:

The contents of the figures were reanalyzed and checked against the text of the manuscript.

Brief information was added in the legend of Figures 2-3 regarding the clinical impact of the molecular and functional relationships shown.

Comment 8:

please be sure the figure 2 is relevant to your study. 

Response 8:

The contents of the figure have been analyzed.
The molecular description of the development of opioid addiction, as well as the hyperactivity of the noradrenergic system during emergency cessation of an opioid agonist are included in the text along with the postulated pathogenesis of opioid use disorder (OUD).

Comment 9: 

please mention if chatbots are utilized for the preparation of the manuscripts.

Response 9: 

No chat-bots were used during the work on the manuscript and the process of selection and analysis of the collected materials.
On the other hand, a translator “DeepL” using artificial intelligence was used locally in the process of writing the text, while it was used only to translate the authors' own work from the mother tongue into English in case of linguistic doubts.

Once again, we sincerely thank you for the detailed review and comments sent regarding our work.
We hope that the changes made and doubts clarified are sufficient, and we are open to further suggestions.

Round 2

Reviewer 3 Report

Comments and Suggestions for Authors

Accept in present form